# Perioperative Fluid Management in Colorectal Surgery: Institutional Approach to Standardized Practice

**DOI:** 10.3390/jcm13030801

**Published:** 2024-01-30

**Authors:** Philip Deslarzes, Jonas Jurt, David W. Larson, Catherine Blanc, Martin Hübner, Fabian Grass

**Affiliations:** 1Department of Visceral Surgery, Lausanne University Hospital CHUV, University of Lausanne (UNIL), 1005 Lausanne, Switzerland; philip.deslarzes@chuv.ch (P.D.); jonas.jurt@chuv.ch (J.J.); martin.hubner@chuv.ch (M.H.); 2Division of Colon and Rectal Surgery, Department of Surgery, Mayo Clinic, 200 First Street SW, Rochester, MN 55905, USA; larson.david2@mayo.edu; 3Department of Anesthesiology, Lausanne University Hospital CHUV, University of Lausanne (UNIL), 1005 Lausanne, Switzerland; catherine.blanc@chuv.ch

**Keywords:** perioperative, enhanced recovery, fluid management, guidance

## Abstract

The present review discusses restrictive perioperative fluid protocols within enhanced recovery after surgery (ERAS) pathways. Standardized definitions of a restrictive or liberal fluid regimen are lacking since they depend on conflicting evidence, institutional protocols, and personal preferences. Challenges related to restrictive fluid protocols are related to proper patient selection within standardized ERAS protocols. On the other hand, invasive goal-directed fluid therapy (GDFT) is reserved for more challenging disease presentations and polymorbid and frail patients. While the perfusion rate (mL/kg/h) appears less predictive for postoperative outcomes, the authors identified critical thresholds related to total intravenous fluids and weight gain. These thresholds are discussed within the available evidence. The authors aim to introduce their institutional approach to standardized practice.

## 1. Introduction

Over the last 20 years, fluid management has been increasingly recognized as a sensitive and modifiable parameter of perioperative care, directly affecting postoperative outcomes [1,2,3]. However, the optimal amount of perioperative fluid administration is controversial, and standardized definitions of a restrictive or liberal regimen are lacking due to conflicting evidence, institutional protocols, and personal preferences [4,5]. In line with these findings, a recent meta-analysis revealed various intra- and postoperative fluid volumes [6]. 

On the one hand, peri- and postoperative fluids are essential to maintain adequate organ perfusion and tissue fluid homeostasis [7]. An overly restrictive approach may lead to hypotension and decreased organ perfusion, ultimately associated with acute kidney injury (AKI) [4]. Furthermore, perioperative organ injury due to both inflammation and ischemia (due to a demand–supply mismatch) represents a potential hazard, thus needing preventive measures and close perioperative monitoring [8]. Enhanced recovery after surgery (ERAS) pathways aim to decrease the physiological surgical stress response represented by a state of insulin resistance [9]. Several measures, including preoperative carbohydrate loading, perioperative feeding strategies, minimally invasive surgery, and early resumption of a normal diet help to modulate the stress response, promote insulin sensitivity, and attenuate the breakdown of protein. Further consequences related to decreased organ perfusion due to an overly restrictive approach may be cardiovascular dysfunction (perioperative myocardial ischemia due to tachycardia, hypotension, hypoxia, or anemia), neurological complications (including confusional states or delirium), and intestinal dysfunction (including splanchnic or anastomotic hypoperfusion), which may be exacerbated by an excessive use of vasopressors [10,11]. 

On the other hand, fluid overload may result in harmful “third space” weight gain, associated with higher rates of pulmonary complications, postoperative ileus, altered mental status, and edema-related anastomotic complications, thus impeding postoperative recovery [12,13,14,15,16]. Furthermore, an excessive extracellular fluid volume may lead to abdominal compartment syndrome, which by itself may trigger adverse physiologic effects such as respiratory failure and renal failure [17]. In light of these findings, definitions must be set to guide clinical practice.

In the setting of established ERAS pathways, the authors’ institutions attempted to identify “safety” fluid thresholds for colorectal resections [13,18,19]. The present review aims to define optimal fluid management, provide an overview of suggested thresholds, and discuss this institutional practice in the light of available evidence.

## 2. What Is Optimal Fluid Management?

Optimal fluid management implies a normovolemic state during and beyond the surgical procedure without fluid management-related complications due to overly restrictive or generous fluid administration, least possible postoperative weight gain, and prompt functional recovery. Whether a specific patient can be managed by noninvasive monitoring and according to a “zero fluid” approach as suggested by the ERAS guidelines mainly depends on the disease presentation, physiological state at the time of surgery, comorbidities, and patient frailty [2]. A euvolemic, otherwise healthy patient without significant comorbidities warranting close surveillance going into elective, minimally invasive surgery is thus eligible for a standardized, restrictive fluid strategy, considering the physiologic principles of euvolemia [5]. On the other hand, patients at risk presenting with an impaired physical condition and distress due to a more acute or emergent disease presentation should benefit from invasive monitoring techniques and be treated within a more liberal strategy according to their physiologic reactions to surgery in a non-elective, acute setting [6]. This is even more important given the fact that these fragile patients are prone to postoperative morbidity and are not eligible for a simplified restrictive approach. On the contrary, management of these patients implies several critical perioperative assessments, including an evaluation of fluid responsiveness triggering, if appropriate, the administration of fluid boluses to increase stroke volume [20]. Of note, such a protocol does not necessarily need hemodynamic monitoring devices for reliable prediction but can also be carried out using echography after a passive leg raising test or by inferior vena cava evaluation, both in mechanically ventilated and spontaneously breathing patients [21,22,23]. In line with these basic principles, both authors’ institutions aimed to standardize fluid management over the last years to implement preset thresholds related to IV fluids and weight gain as red flags for guidance in clinical practice.

### Definition of a Restrictive versus Liberal Approach

To date, there is no standardized definition of restrictive fluid therapy. The Enhanced Recovery After Surgery (ERAS) guidelines recommend aiming for a “zero fluid” balance and euvolemia intraoperatively and during the first postoperative days in patients undergoing elective colorectal resections [24,25]. Pre-operatively, carbohydrate loading and unrestricted access to clear fluids until 2 h before anesthesia induction help maintain fluid homeostasis and initiate surgery in a euvolemic, physiological state. Intraoperatively, a basal rate of crystalloid solution of <4 mL/kg/h is recommended [24,26]. This approach has been considered “restrictive”; however, its interpretation and application in clinical practice remain vague and subjective. Patients requiring goal-directed fluid therapy (GDFT) should receive boluses to maintain the cardiac stroke volume and, hence, central normovolemia [6]. However, recent guidance reserves a GDFT approach for high-risk patients (e.g., frailty and cardiopulmonary dysfunction) and high-risk procedures (e.g., emergent setting and disease-related distress) with large intravascular fluid loss [25,27,28]. Postoperatively, both early IV fluid lock and resumption of liquids and solids allow for adherence to the natural process of fluid homeostasis according to individual needs [29]. 

In a recent meta-analysis including 18 randomized controlled trials, the median intraoperative fluid administrated in the restrictive group was 1930 mL (interquartile range (IQR): 1480–2470 mL) compared to 3880 mL (IQR: 3000–4400 mL) in the liberal group [30]. On postoperative day 1, the median volume of intravenous fluids was 2340 mL (IQR 1640–3530 mL) versus 4350 mL (3100–5330 mL), respectively. However, important differences were observed among individual trials regarding total fluid volumes in the restrictive and liberal groups [30,31]. Consequently, a liberal approach in a specific trial could be equivalent to a restrictive approach in another trial [30,32]. While the concept of fluid restriction outside high-risk patients and procedures is widely accepted, “safety” thresholds may be valuable adjuncts and serve as red flags for clinical guidance during anesthesia and postoperative surveillance. Several randomized controlled trials compared both approaches (restrictive vs. liberal) and reported on fluid-related thresholds and postoperative complications, as summarized in Table 1.

Table 1 provides an overview of published RCTs comparing restrictive and liberal groups.

In a former meta-analysis, Varadhan et al. suggested stratifying fluid regimens of the perioperative day into restrictive (<1750 mL/d), balanced (1750–2750 mL/d), and liberal (>2750 mL/d) [32]. The balanced fluid range was calculated to compensate for the daily physiological water loss for an average human in a homeostatic state, estimated between 25–35 mL/kg [46,47]. This volume is supposed to replace the perioperative body water loss to approach a zero fluid balance. Interestingly, this upper cut-off of 2.7 L was independently confirmed by an institutional series of the Mayo Clinic [13].

## 3. Impact of Fluid Overload on Postoperative Complications

A considerable weight gain of >6 kg after elective colorectal surgery has been observed in several studies, requiring close postoperative surveillance to prevent associated complications, especially in fragile patients prone to pulmonary complications [33]. However, fluid management in these fragile patients represents a particular challenge given that they are at increased risk of experiencing postoperative morbidity. This impedes uncritical assumptions of cause (fluid overload) and effect (complications) patterns. While some of the data suggest a modest correlation between total perioperative IV fluid administration and weight gain [48,49], a dose–response correlation with consequent increased complication rates was observed by others [33,50]. Despite the seemingly easy-to-perform weight measurements in the postoperative period, postoperative weight is reported in only 50% of randomized controlled studies [30], Table 1.

Fluid overload induces prolonged gastric emptying [5], which, together with bowel edema and interstitial third space fluids, causes postoperative ileus (POI). The series of both our institutions confirmed an independent effect of fluid overload and weight gain on POI occurrence [18,51]. These findings were confirmed by others and independently validated [52,53,54]. Furthermore, similar associations were observed in the setting of ostomy procedures [55,56].

Pulmonary complications after surgery are a major concern, with an occurrence of up to 23% [12,57]. Fluid overload of the interstitial space triggers pulmonary edema, especially in patients with impaired cardiac function [57,58]. A significant decrease in mean blood saturation on the second night after surgery was observed in patients within the liberal fluid administration group; however, there was no increased morbidity in this study [37]. However, the results are conflicting, and cause–effect patterns are hard to establish in fragile patients with cardiopulmonary impairment. Several studies, including an institutional series, revealed that fluid overload and weight gain are associated with an increased risk of pulmonary complications [12,50,59]. 

### Impact of Fluid Management on Renal Function 

While perioperative hypotension may impact on several organs, a major concern of overly restrictive perioperative fluid administration is the development of AKI. The evidence is conflicting. A meta-analysis revealed a higher AKI rate in the restrictive group [30]. Further data suggest that even a minor increase in creatinine levels could increase in-hospital mortality in non-cardiac surgical patients [60]. However, no cause–effect patterns could be established due to its retrospective design. Myles et al. published a large multicentric randomized controlled landmark trial comparing restrictive versus liberal fluid administration in major abdominal surgery [4]. In their study, the restrictive approach had no impact on disability-free survival but was associated with a statistically significant AKI increase (8.6% vs. 5% in the restrictive and liberal groups, respectively). Notably, around 50% of patients in this trial were not treated according to the ERAS principles, impeding uncritical extrapolation of the results to the setting of our institutions offering care within longstanding, established, and standardized ERAS pathways [61,62]. A sizeable institutional series of elective patients revealed a low AKI rate of 2.5% according to loss of kidney function and end-stage kidney disease (RIFLE) criteria [63]. In another series of our group, an intraoperative fluid range defined as “balanced” (300 mL–2700 mL) was associated with the lowest rate of POI and a prolonged length of stay but not AKI [13]. Restrictive fluid management during elective colorectal resections appears safe if carried out within standardized pathways and it is supported by respective societies [24,25,64]. 

## 4. Fluid Management in the Perioperative Period: Which Indicators 

Intraoperative oliguria occurring in isolation should not trigger fluid boluses since the predictive value for postoperative AKI appears low [65]. An institutional series of the Mayo Clinic revealed that a certain degree of postoperative hypotension in up to 10% of patients may persist for less than 20 h without negatively impacting AKI occurrence, which affected <3% [66]. There is a broad consensus that a permissive attitude to physiologic oliguria due to renal vasoconstriction can be adopted in the elective ERAS setting, providing no established cause exists [25]. Based on the available information, intraoperative fluid management should be protocolized to determine an underlying physiologic problem requiring reversal [67]. Standard monitoring integrating clinical data is thus likely sufficient in low-risk procedures, combining maintenance fluids at a low rate of < 4 ml/kg/h in the intraoperative and early postoperative period in the post-anesthesia care unit. Outside this low-risk setting and depending on the surgical risk, GDFT, including advanced hemodynamic monitoring devices, should be used as valuable adjuncts in higher-risk patients or procedures, triggering fluid administration if a decreased cardiac output or surrogates are suspected [68,69]. 

## 5. Summary of Institutional Thresholds and Practice Guidance

Based on the above discussed evidence and considering a 7-year experience in ERAS care in both authors’ institutions at that time, our groups aimed not only to focus on established, evidence-based perioperative ERAS care but also to standardize fluid management [19]. The need to improve perioperative fluid management standards in our institutions was motivated by the rather low compliance with guidelines, despite growing ERAS experience [19]. Importantly, the aim was not to set inflexible, dogmatic thresholds but to help with guidance in clinical practice. Restrictive fluid management through a zero-balance practice in elective surgery represents one puzzle piece in a comprehensive care pathway aiming to maintain a physiologic state throughout the perioperative period, significantly impacting postoperative recovery [5]. 

In total, 11 cohort studies of the authors’ institutions described fluid management-related thresholds, as summarized in Table 2. 

The thresholds are displayed with their respective impact on specific outcomes or clinical consequences. Three papers from the Lausanne group tried to identify thresholds through receiver operating characteristics (ROC) curves in different surgical settings: minimally invasive surgery [50], open surgery [73], and lastly, surgery for urgent indications [72]. Interestingly, the thresholds did not differ significantly across the different settings. The Mayo group analyzed an independent large dataset of elective colorectal surgeries with a focus on POI, prolonged LOS, and AKI, which were plotted against the rate of intraoperative Ringer lactate (RL) infusion (mL/kg/h) and total intraoperative volume [13]. Total intraoperative RL ≥2.7 L was independently associated with POI and prolonged LOS, but not AKI. Of note, the infusion rate (ml/kg/h) was not retained as a superior predictive tool. Further work focused on patients undergoing major surgery and needing postoperative surveillance in an intermediate care unit [48]. In this particularly vulnerable subgroup of patients, the fluid balance and weight course showed only a modest correlation. Both institutions further focused on POI in their analyses and found comparable results, with a strong correlation of fluid overload and POI in patients undergoing major surgery [18] and in patients undergoing loop ileostomy closure [56]. In the largest dataset of the Mayo group with over 7000 patients, early AKI was very uncommon within the institutional ERP (2.5%), and long-term sequelae were exceptionally low [63]. Interestingly, AKI patients received higher amounts of POD 0 fluids and had increased postoperative weight gain at POD 2. A further study of the Lausanne group revealed a protective effect of high compliance with the ERAS protocol to prevent postoperative pulmonary complications [12]. A threshold of 4 kg at POD 2 appeared to be critical in this setting. Finally, both author groups showed increasing interest in short stay processes in recent years, and excess intraoperative fluids of >3 L turned out to impede early discharge and thus an outpatient strategy [70,71].

Taking the above summarized evidence together, a threshold of 3000 mL intraoperatively serves presently as a red flag in daily clinical practice in both authors’ institutions. In addition to the mere focus on IV fluids, weight gain at postoperative day 2 turned out to be a valid surrogate for fluid overload [18]. 

Besides IV fluid management, several further ERAS care items help to maintain tissue homeostasis and an euvolemic state [24]. Preoperative carbohydrate loading helps to attenuate the catabolic response through a reduction of insulin resistance in response to surgery [74]. Clear fluids can be safely ingested until 2 h before surgery, whereas 6 h fasting for solid food is sufficient [75]. While there is growing evidence in favor of combined mechanical and oral antibiotic bowel preparation, mechanical bowel preparation alone may lead to preoperative dehydration and electrolyte imbalances and should thus be avoided [76]. Postoperatively, early oral nutrition is advocated and has proven its benefits by several meta-analyses and has been endorsed by different nutritional societies [64]. Finally, early mobilization of at least 6 h per day is of utmost importance and helps to prevent muscle loss and to promote functional recovery due to a direct prokinetic effect on the intestines [77]. Figure 1 summarizes the pre, intra, and postoperative measures within the institutions’ standardized ERAS protocol.

## 6. Implications in Daily Clinical Practice

The fast track concept that eventually led to standardized ERAS pathways was introduced 25 years ago by Henrik Kehlet and helped to simplify patient management by targeting the quality and speed of postoperative recovery [78]. Standardization of care is a way to facilitate patient management and improve a multidisciplinary team approach [79]. This holds true for surgical technique, but also intraoperative management and patient care in the ward. Postoperative care protocols with predefined care maps simplify the workflow, especially for frequently performed procedures. Perioperative fluid management represents a key element of ERAS care.

ERAS guidelines suggest aiming for a zero fluid balance for elective colorectal resections, while GDFT should be reserved for high-risk patients and procedures [24,25]. The use of vasopressors is advocated when fluid boluses fail to improve the stroke volume in order to prevent fluid overload [80]. The thresholds described in the present study and used in the authors’ institutions cannot replace careful individual risk-stratification in every patient before surgery. However, in the authors’ experience, they help with raising awareness among both surgeons and anesthesiologists to discuss fluid management during and after the procedure. Furthermore, a weight gain threshold of 2.5 kg at POD 2 serves as a useful point of reference in the surgical ward. Postoperative body weight is easy to assess and helps to timely launch counterregulatory measures [48,81]. In patients who exceed the threshold, subsequent fluid restriction, diuretics, and the promotion of mobilization can be initiated [50].

## 7. Conclusions

In conclusion, our practice of restrictive fluid management is based on institutional thresholds to help guide clinical practice, aiming to prevent deleterious fluid overload-related adverse outcomes.

## Figures and Tables

**Figure 1 jcm-13-00801-f001:**
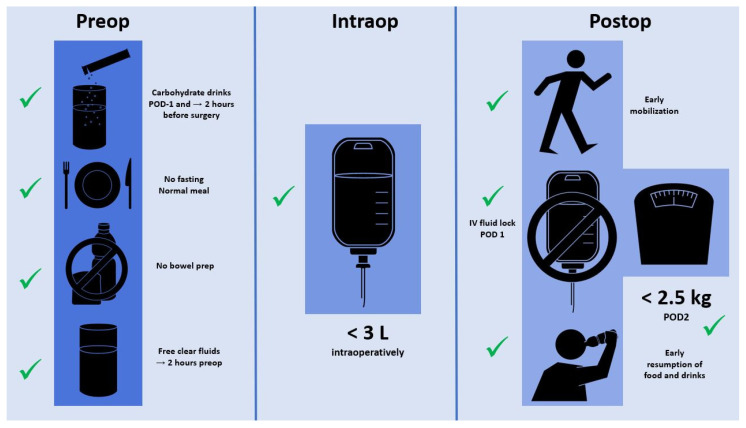
Schematic representation of fluid management-related recommendations within the authors’ institutional ERAS pathways.

**Table 1 jcm-13-00801-t001:** Randomized controlled trials comparing restrictive and liberal fluid regimens.

Study (Year)	Surgery	N	Total Fluids	IV Fluid Management	mL/kg/h	Weight Day 2 (∆, kg)	Outcomes Restrictive Group
Lobo 2002 [5]	Elective CS (cancer)	10 (R)	11.6L (IV + oral)	3000 (POD 0)	NA	0	↓ LOS, ↓ gastric emptying
		10 (L)	18L (IV + oral)	5700 (POD 0)		3	↓ time to stool
Brandstrup 2003 [33]	Elective CRS	69 (R)	3.8L (IV + oral POD 0)	2700 (POD 0)	NA	1	↓ cardiopulmonary + tissue-healing complications
		72 (L)	6.2L (IV + oral POD 0)	5400 (POD 0)		3.8	
Nisanevich 2005 [34]	Major abdominal surgery	77 (R)	NA	1400 (IO), 2200 (POD1)	4 RL (IO)	0.5 (POD 1)	↓ LOS, ↓ time to flatus/stool
		75(L)		3900 (IO), 2000 (POD1)	12 RL (IO)	1.9 (POD 1)	
Kabon 2005 [35]	Elective CS	124 (R)	NA	2500 (IO)	8–10 RL (IO)	NA	→ SSI, nausea
		129 (L)		3900 (IO)	16–18 RL (IO)		
MacKay 2006 [36]	Elective CS	39 (R)	NA	2000 (IO), 2000 (POD1)	NA	NA	→ time to flatus/stool, LOS
		31 (L)		2750 (IO), 2600 (POD1)			
Holte 2007 [37]	Elective CS	16 (R)	1600 (POD 0)	1140 (IO)	5–7 RL	0.8	→ complications, time to stool, LOS
		16 (L)	5100 (POD 0)	3900 (IO)	18 RL	2.9	
Muller 2009 [38]	Elective CS	76 (R)	2700 (POD 0)	1900 (IO)	5 RL (IO)	NA	↓ complications, ↓ LOS
		75 (L)	5200 (POD 0)	3000 (IO)	10 RL (IO)		
Aguilar-Nascimento 2009 [39]	Major abdominal surgery	28 (R)	9.2 L	4400 (IO)	17	NA	↓ LOS, ↓ pulmonary complications
		33 (L)	11.7 L	5400 (IO)	20		
Futier 2010 [40]	Major abdominal surgery	36 (R GDT)	NA	3400 (IO)	7.7	NA	↑ complications (leak, sepsis)
		5600 (IO)		12.2	24 (C GDT)		
Abraham-Nordling 2012 [41]	Elective CRS	79 (R)	NA	3100 (POD0)	NA	0.8 2.9	↓ overall complications,
		82 (L)		5800 (POD0)			→ LOS, leak, AKI,
							↑ cardiac complications
Kaylan 2013 [42]	Elective CRS	121 (R)	NA	1000 (IO), 1900 (POD0)	5–7 (L, IO)	−1.4	→ major complications, LOS, mortality
		119 (L)		2000 (IO), 3300 (POD0)		1.3	
Hong-Ying 2014 [43]	Elective CRS (cancer)	96 (R)	NA	1600 (IO)	NA	0.9	↓ overall complications,
		89 (L)		3100 (IO)		2.8	↑ cardiac complications
Phan 2014 [44]	Elective CRS	50 (R)	NA	1500 (IO)	5 (both groups)	NA	→LOS, minor/major complications
		50 (L)		2100 (IO)			
Gomez-Izquierdo 2017 [45]	Elective CRS	64 (GDT) 64 (L)	NA	1500 (IO) 2400 (IO)	6 (GDT) 12	0.6 0	→ ileus, LOS, surgical and medical complications
Myles 2018 [4]	Major abdominal surgery	1490 (R)	NA	1700 (IO), 3700 (24 h)	NA	NA	↑ AKI
		1493 (L)		3000 (IO), 6100 (24 h)			→ sepsis, mortality

IV—intravenous, CS—colon surgery, CRS—colorectal surgery, R—restrictive, L—liberal, GDT—goal-directed therapy, NA—not available, LOS—length of stay, POD—postoperative day, IO—intraoperative, AKI—acute kidney injury. Total fluids relate to the total LOS unless specified otherwise. Arrow down: decreased, arrow up: increased, regular arrow: same.

**Table 2 jcm-13-00801-t002:** Fluid thresholds and related outcomes within the authors’ institutions.

Study (Year)	Cohort	N	Critical Fluid-Related Threshold	Outcome Related to Fluid Overload
Abd El Aziz 2022 [13]	Elective CRS	2900	300–2700 mL (IO)	↑ POI, ↑ LOS, ↑ AKI
Grass 2022 [70]	Elective CRS	5′398	3000 mL (IO)	Impeded outpatient strategy in selected patients
Butti 2020 [48]	Major abdominal surgery + IMC stay	111	3 kg (POD 2)	Prolonged IMC stay
Grass 2020 [18]	Elective CRS	4205	3000 mL	↑ POI
			2.5 kg (POD 2)	
Grass 2020 [71]	Elective CRS	5122	3000 mL (IO)	Prolonged LOS > 48 h
Grass 2020 [72]	Urgent colectomy	224	3000 mL (POD 0),	↑ overall complications
			2.3 kg (POD 2)	
Grass 2019 [63]	Elective CRS	7103	3800 mL	↑ AKI
Hübner 2019 [50]	Laparoscopic CRS	580	3000 mL (colon)	↑ overall, major, respiratory complications
			4000 mL (rectum)	
Grass 2019 [56]	Loop ileostomy closure	238	1700 mL (POD 0)	↑ POI
			1.2 kg (POD 2)	
Pache 2019 [73]	Open CRS	121	3500 kg (POD 0)	↑ overall, respiratory complications, prolonged LOS
			3.5 kg (POD 2)	
Jurt 2018 [12]	Elective CRS	1298	4 kg (POD 2)	↑ respiratory complications

IV—intravenous, CS—colon surgery, CRS—colorectal surgery, LOS—length of stay, POD—postoperative day, IO—intraoperative, AKI—acute kidney injury, POI—postoperative ileus. Arrow down: decreased, arrow up: increased, regular arrow: same.

## Data Availability

Please contact the authors for specific data.

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
