# Peer review of "Perioperative Fluid Management in Colorectal Surgery: Institutional Approach to Standardized Practice"

_jcm, 2024, doi:10.3390/jcm13030801_

Round 1

Reviewer 1 Report

Comments and Suggestions for Authors

I thank the editors for the opportunity to review the manuscript by Deslarzes et al., and congratulate the authors for their short and concise overview on preoperative fluid management in colorectal surgery. They present an easy-to-read and important contribution to the ongoing discussion on administration of fluids perioperatively. I have the following points I invite the authors to address:

- I would argue that the main aim of the author's paper is to present their idea and take on how to give fluids perioperatively based on their extensive preliminary work. However, this paragraph is the shortest, and recommendations are mainly presented in a figure. I recommend to expand this section significantly.

- As a surgeon, the most important factor for me is the impact on postoperative complications. The authors did describe these in detail, however, there is often confusion concerning the interaction of fragile/chronically ill patients, fluid management and postoperative morbidity. I suspect that fragile patients are prone to receive more fluids, however, their fragility impacts postoperative morbidity itself. I invite the authors to distinguish the impact of multi-morbidity and fluid management.

Author Response

Thank you for your kind evaluation and thoughtful comments.

  • We substantially expanded the discussion of our own work in the field as suggested (paragraph 5).
  • Regarding your 2nd comment, we added two statements to emphasize this critical observation (paragraph 2, lines 73-75, paragraph 3, lines 126-128).

Reviewer 2 Report

Comments and Suggestions for Authors

I read with great interest the manuscript by Deslarzes et al. on the perioperative fluid management in colorectal surgery. The review is interesting. However, there are several issues that need to be addressed.

- Line 33-35. Fluid restriction and consequent hypotension is not only associated with AKI, but also with a variety of organ damages related to hypoperfusion. Please specify and add adequate references.

- Line 40. Do authors mean ERAS protocols? I would suggest to use this definition as it is more frequently used in literature.

- Line 55. Please replace "risk patients" with "patients at risk".

- Line 55-58. In this sentence, I believe authors should be more precise and explain the rationale of using hemodynamic monitoring. In fact, patients at risk should be carefully assessed to verify whether they are fluid responders, so that they would benefit from a fluid bolus in terms of an increase of stroke volume (doi: 10.1007/s00134-022-06900-0). Moreover, this assessment can be done not only with hemodynamic monitoring, but also using echocardiography after a passive leg raising test (doi: 10.1007/s00134-015-4134-1) or by inferior vena cava evaluation both in mechanically ventilated (doi: 10.1186/s40635-023-00529-z) and spontaneously breathing patients (doi: 10.1186/s40635-023-00505-7). Please discuss and add these 4 references.

- Line 71. What do you mean by invasive?

- Line 123. As mentioned before, I believe authors should explore more the role of hypotension on the function of other organs.

- Line 158. In this paragraph the authors show the protocol developed for their institution. However, the protocol is not explained (Figure 1: Preoperative fasting is not specified. Do patients eat freely before surgery? What do authors mean by early mobilization? How many days? etc.). Did authors test this protocol? In table 2, I believe authors should perform a qualitative synthesis of the data provided, otherwise this table does not provide any useful information to the review. Please modify or remove the paragraph.

Author Response

Thank you very much for your thoughtful coments. Please find below answers and in the manuscript respective specifications to each point.

  • comment 1: we specified as suggested that a variety of other organs can be damaged and respective references (lines 36-48).
  • comment 2: we switched to ERAS protocol throughout
  • comment 3: replaced by "patients at risk"
  • comment 4: we gladly added this important information and respective pertinent references (paragraph 2, lines 71-81)
  • comment 5: we deleted "invasive" and apologize for the typo
  • comment 6: Please see above, we substantially expanded on this important point (please also see lines 147-148).
  • comment 7: We added an extensive paragraph to discuss the institutional ERAS protocol and all raised queries (please see paragraph 5 and the new paragraph 6)

Round 2

Reviewer 2 Report

Comments and Suggestions for Authors

The authors successfully addressed all the comments provided. I believe the manuscript can be accepted in its present form.